# Engineered Liposomes Protect Immortalized Immune Cells from Cytolysins Secreted by Group A and Group G Streptococci [note 1]

**DOI:** 10.3390/cells11010166

**Published:** 2022-01-05

**Authors:** Hervé Besançon, Yu Larpin, Viktoria S. Babiychuk, René Köffel, Eduard B. Babiychuk

**Affiliations:** Institute of Anatomy, Faculty of Medicine, University of Bern, 3012 Bern, Switzerland; yu.larpin@ana.unibe.ch (Y.L.); viktoria.babiychuk@ana.unibe.ch (V.S.B.); rene.koeffel@ana.unibe.ch (R.K.)

**Keywords:** antibiotic resistance, bacterial infection, *Streptococcus*, toxin, liposome, nanotrap, immune cells, antivirulence

## Abstract

The increasing antibiotic resistance of bacterial pathogens fosters the development of alternative, non-antibiotic treatments. Antivirulence therapy, which is neither bacteriostatic nor bactericidal, acts by depriving bacterial pathogens of their virulence factors. To establish a successful infection, many bacterial pathogens secrete exotoxins/cytolysins that perforate the host cell plasma membrane. Recently developed liposomal nanotraps, mimicking the outer layer of the targeted cell membranes, serve as decoys for exotoxins, thus diverting them from attacking host cells. In this study, we develop a liposomal nanotrap formulation that is capable of protecting immortalized immune cells from the whole palette of cytolysins secreted by *Streptococcus pyogenes* and *Streptococcus dysgalactiae* subsp. *equisimilis*—important human pathogens that can cause life-threatening bacteremia. We show that the mixture of cholesterol-containing liposomes with liposomes composed exclusively of phospholipids is protective against the combined action of all streptococcal exotoxins. Our findings pave the way for further development of liposomal antivirulence therapy in order to provide more efficient treatment of bacterial infections, including those caused by antibiotic resistant pathogens.

## 1. Introduction

Antibiotic resistance, in combination with a longer living and increasingly immunocompromised population, is a global health threat. Whereas pathogens continue to adapt and evade the currently available therapies, new antibiotics are lacking in the pharmaceutical pipelines, as they are difficult and expensive to develop and commercialize [1].

Common guidelines to fight antibiotic resistance focus on the development of alternative treatments [2]. One promising approach is antivirulence therapy, which inhibits virulence factors—molecules that are produced by pathogens in order to establish a persistent infection [3]. Antivirulence drugs strip the pathogen of its virulence properties but do not directly kill it, thereby reducing selective pressure to develop resistance mechanisms [4]. Antivirulence drugs promote bacterial clearance by the immune system and can be used as joined therapeutic agents with antibiotics.

Virulence factors include exotoxins, secreted by bacterial pathogens to harm host cells, and allow for bacterial survival and efficient replication in a hostile environment. Secreted exotoxins/cytolysins, more specifically pore-forming toxins (PFTs), are common virulence factors of both Gram-positive and Gram-negative bacteria [5,6,7,8]. Pathogens producing PFTs include *Streptococcus* spp., *Clostridium* spp., *Listeria* spp., and *Bacillus* spp. [5]. Secreted as soluble monomers, PFTs bind to and perforate the plasma membrane of the host cells by forming oligomeric transmembrane pores. Plasmalemmal perforation results in a loss of cytoplasmic content and ion disbalance, disturbs the proper functioning of the cell, triggers stress pathways, and can result in inflammation or activation of programmed cell death [9]. PFTs damage tissues and organs; facilitate bacterial dissemination; prevent killing by phagocytic pathway evasion; disable host immune defenses; and, in some instances, prompt highly damaging immune over-responses, leading to toxic shock syndromes [10,11].

*Streptococcus pyogenes* (Group A *Streptococcus*, GAS) and *Streptococcus dysgalactiae* subspecies *equisimilis* (Group G *Streptococcus*, GGS) are widespread Gram-positive pathogens. They are throat, vaginal tract, and skin colonizers [12,13]. Infections by GAS and GGS can result in a wide spectrum of diseases, ranging from pharyngitis to life-threatening bacteremia or streptococcal shock syndrome [12,14,15]. GAS ranks among the top 10 causes of infection mortality worldwide [15,16,17].

GAS and GGS secrete two cytolysins, Streptolysin O (SLO) and Streptolysin S (SLS). SLO is a member of the cholesterol-dependent cytolysins (CDCs) family [18]. SLO monomers bind to cholesterol-containing membranes and oligomerize to create a pore [19]. SLS is a small peptide of the thiazole/oxazole-modified microcins class [20]. We showed previously that SLS binds with a low affinity to plasmalemmal phosphatidylcholine and sphingomyelin, but its exact mode-of-action is not yet fully clarified [21].

To establish an infection, invading pathogens need to fight off the host immune cells. If unharmed, innate immune cells such as macrophages, as first responders, can kill microorganisms by phagocytosis, serve as antigen-presenting cells, and produce mediators that trigger local inflammation and recruit adaptive immune cells such as T and B lymphocytes [22]. Immune cells use various mechanisms to repair holes in their membrane and counteract ion disbalance, but if overloaded with toxins, the cells end up dying [23]. If immune cells are killed or sufficiently injured, the pathogen establishes itself. Macrophage depletion, especially early in the infection course, severely increases GAS proliferation and the death rate in mice [22]. It is paramount to protect immune cells against GAS and GGS exotoxins that have been shown to promote apoptosis in various immune cell types [24,25].

PFTs, irrespective of their mode-of-action, have to interact with the targeted cell membrane, which acts as a barrier between the cytoplasm and the extracellular milieu. PFTs bind to specific targets on the membrane, often cholesterol or phospholipids [18,26,27]. Using these interactions, it is possible to give the toxin a decoy target to divert it from attacking the host cells. We and others have developed nanotraps to capture and neutralize bacterial exotoxins, either as nanosponges composed of cell-derived lipid bilayers wrapped around a polymeric core or liposomes composed of naturally occurring purified lipids forming one or more lipid bilayers [21,28,29].

Liposomal nanotraps can neutralize exotoxins that are produced by pathogens from different families, and rely on different mode-of-action, such as CDCs, SLS, phospholipase C, or α-hemolysin [21,29].

In vivo liposomal nanotraps rescued infected mice from deadly bacteremia and pneumonia induced by *Streptococcus pneumoniae* and *Staphylococcus aureus* [29]. This approach also proved successful in a first-in-human study in patients with severe pneumonia caused by *Streptococcus pneumoniae*. Used as an add-on therapy to antibiotics, liposomal nanotraps shortened the time of infection [30]. Pneumococcal toxins were successfully neutralized and the liposomal agent was determined to be safe and well tolerated [30].

Here, we use purified bacterial supernatants produced by GAS and GGS bacteria to challenge a monocytic (THP-1), a lymphoid T (Jurkat) and a lymphoid B (Raji) cell lines with streptococcal exotoxins. We show that cholesterol-containing liposomes, unsaturated phosphatidylcholine-liposomes, or a combination of both liposomes protect the immune cells. We also highlight the differences in the sensitivity of different cell lines to streptococcal toxins.

## 2. Materials and Methods

### 2.1. Bacterial Cultures

Bacterial culture supernatants were prepared from *Streptococcus pyogenes* strains 19165 (ATCC, Manassas, VA, USA), 50362 (clinical isolate from a biopsy), *Streptococcus dysgalactiae* subspecies *equisimilis* 12394 (ATCC, Manassas, VA, USA), and 5804 (clinical isolate, septic arthritis) [31]. Clinical isolates were kindly provided by Parham Sendi and Lucy J. Hathaway (Institute for Infectious Diseases, University of Bern, Bern, Switzerland). Bacteria were grown overnight in brain heart infusion (BHI) (Sigma-Aldrich, Saint-Louis, MO, USA) with 10% fetal bovine serum (FBS) (Seraglob, Schaffhausen, Switzerland) at 37 °C. The culture was diluted 1:100 in BHI-FBS (10%) and incubated at 37 °C until reaching an OD_540_ of 1. Bacterial cultures were centrifuged at 4 °C for 40 min at 4000 rpm. Culture supernatants were filtered through a 0.45 μm filter (Sarstedt, Nümbrecht, Germany), pH adjusted to 7, aliquoted, and stored at −80 °C until further use.

### 2.2. Cell Cultures

THP-1 (TIB-202), Jurkat (TIB-152), and Raji (CCL-86) cells were purchased from ATCC (Manassas, VA, USA). The cells were maintained in a RPMI1640 medium (Gibco, Life Technologies, Carlsbad, CA, USA) supplemented with 10% FBS and 1% penicillin-streptomycin (Gibco, Life Technologies, Carlsbad, CA, USA) in 5% CO_2_ at 37 °C.

### 2.3. Liposomal Nanotraps

Egg sphingomyelin (Sm), 18:1/18:1 phosphatidylcholine (PC), and cholesterol (Ch) were purchased from Avanti Polar Lipids (Alabaster, AL, USA) in powder form. Liposomes were produced by sonication (20 min on ice; 5 × 10% cycles at maximal power (Bandelin Sonoplus, Berlin, Germany) and kept at 4 °C until further use, as previously described [29]. The diameter of the liposomes was measured by NanoSight NS300 (Malvern Panalytical, Malvern, UK) for Ch:Sm-liposomes (108 ± 7 nm) and 18:1/18:1 PC-liposomes (130 ± 9 nm). Liposome amounts correspond to the amount of total lipids used for their preparation.

### 2.4. Cells Survival

A total of 5 × 10^4^ cells were added to air permeable tubes (Sarstedt, Nümbrecht, Germany). For the cytotoxicity assays, bacterial supernatant was serially diluted (step 2, PBS) and added to the mix. For protection experiments, serial dilutions (step 2, PBS) of liposomal nanotraps were added to the cells; the assay was initiated by the addition of a fixed volume of the bacterial supernatant (final reaction volume = 500 μL). The added supernatant volume was determined based on the toxicity assay results and was used either at saturating dose (lethal dose > 90, LD_>90_) or at non-saturating dose (LD_60–90_) to study minor toxin activities. The mixture was incubated for 3 h at 37 °C with 5% CO_2_. The cells were centrifuged 5 min at 1100 rpm, suspended in fresh medium, and incubated in standard conditions for 3 to 5 days. Surviving cells were counted using a CellDrop (DeNovix, Wilmington, DE, USA). The data were normalized to a control incubated with PBS instead of the bacterial supernatant (considered as 0% cell death). A control challenged by the bacterial supernatant without liposomes was added for each assay to verify the expected cytotoxic activity. Alamar blue (Invitrogen, Waltham, MA, USA) cell viability assay and Trypan blue (Gibco, Life Technologies, Carlsbad, CA, USA) live/dead quantification provided results that are identical to those obtained in the cell proliferation protocol used in our study (not shown).

### 2.5. Mass Spectrometry

Precleaned supernatants (100,000 g for 2 h, 625 μL) and Ch:Sm-liposomes (200 μg) were diluted in PBS (5 mL) and incubated for 15 min at 37 °C. Liposomes were pelleted by ultracentrifugation at 100,000 g for 2 h. The pellets were re-suspended in 100 μL of PBS. The samples were heated for 10 min at 95 °C in a sample buffer (0.5 M Tris–HCl pH 6.8; 40% glycerol; 4% SDS; 5 µL/ml bromphenolblue; 5% beta-mercaptoethanol). The proteins were separated and the lipids were discarded by SDS-PAGE separation. The proteins were visualized using Coomassie staining/destaining. The migration band was cut in small cubes (1 mm^3^) and was analyzed by mass spectrometry.

The gel pieces were reduced, alkylated, and digested by trypsin, as described elsewhere [32]. The digests were analyzed by nano-liquid chromatography coupled to tandem mass spectrometry (nLC-MS/MS) with one injection of 5 μL digest and a 40 min acetonitrile peptide separation gradient, as described previously [33].

Spectra interpretation was performed with Easyprot searching against the forward and reversed SwissProt *Streptococcus pyogenes* protein database using fixed modification of carboamidomethylation on Cys, and variable modifications of oxidation on Met, deamidation on Asn/Gln, and acetylation on protein N-Term. Mass error tolerance for parent ions was set to 0.4 Da, with a fragment ion tolerance of 20 ppm, and full trypsin cleavage specificity with two missed cleavages was allowed. Based on reversed database matches, a 1% false discovery rate (FDR) was set for the acceptance of peptide spectrum matches (PSM), peptides, and proteins. Protein identifications were only accepted when two unique peptides fulfilling the 1% FDR criterium were identified.

The experiment was performed three separate times.

### 2.6. Statistical Analysis

Paired *t*-tests were performed in R (https://www.r-project.org, version 4.0.5, accessed on 31 March 2021).

## 3. Results

### 3.1. Sensitivity of Immune Cells to GAS or GGS Supernatants Is Strain and Cell Type Dependent

In our previous work, we established that bacterial culture supernatants obtained from both GAS and GGS strains were highly lytic towards erythrocytes [21].

Here, we examined whether GAS and GGS strains possessed cytolytic and/or cytotoxic activities towards the following nucleated immune cells: THP-1 (monocytic cell line), Jurkat (T lymphoid cell line), and Raji (B lymphoid cell line). In our current study, we tested the bacterial culture supernatants obtained from the GAS and GGS reference strains (GAS ATCC 19165 and GGS ATCC 12394) and from clinically isolated strains (GAS 50362 and GGS 5804).

All immune cells died when challenged with bacterial supernatants, indicating that all GAS and GGS strains that were used in this study possess potent cytolytic/cytotoxic activities (Figure 1).

Both GAS strains were most effective against Jurkat and THP-1 cells, whereas Raji cells were the most resistant towards toxic activities of these strains (Figure 1a,b). In contrast, supernatants obtained from both GGS strains were more active against Raji cells than against THP-1 cells (Figure 1c,d). We have previously shown that GAS strains rely to a greater extent on SLO, whereas GGS preferentially use SLS for their hemolytic activities [21]. Therefore, our data suggest that SLO and SLS differ in their ability to harm different types of immortalized immune cell lines. The most notable difference is that SLO is more toxic towards the monocytic cell line (THP-1 cells) compared to the B-lymphoid cell line (Raji cells), whereas SLS shows a fully reversed cytotoxic pattern.

### 3.2. Cholesterol-Containing Liposomes Fully Neutralize Cytotoxic Activities of GAS Supernatants but Only Partially That of GGS Supernatants

We have previously shown that GGS ATCC 12394 relied on both SLO and SLS for their hemolytic activities [21]. GAS ATCC 19165 also secreted both toxins, but its SLO activity was prevalent [21]. In contrast, the hemolytic activities of clinically isolated strains were almost exclusively dependent on either SLO (GAS 50362) or SLS (GGS 5804) [21].

SLO, a member of the CDC family, binds to cholesterol with a high affinity [21]. Correspondingly, Ch-containing liposomes were capable of selective and complete neutralization of this toxin [21,29].

Therefore, we tested whether the neutralization of SLO by Ch:Sm-liposomes (Ch = 66 mol/%; 100 ng ≈ 885 nM total lipid or 584 nM Ch) protected immune cells against GAS and/or GGS supernatants. Direct binding of SLO present in the GAS 50362 supernatant to Ch:Sm-liposomes was verified by a liposome pull-down assay followed by a mass spectrometry analysis of the resulting pellet (Peptide Match Score Summation (~200) ten-fold higher than nonspecific hits (~20), not shown) [34,35].

In our previous work we showed that cholesterol-dependent cytolysins (SLO and PLY) displayed different kinetics and dynamics of their hemolytic activity compared to SLS [21]. As a result, the total hemolytic activities of individual streptococcal supernatants were not represented by a simple sum of activities (concentrations) of their individual toxins, but displayed more complex time- and amount-dependent behavior. For this reason, in our current experiments, we used the specific cytotoxic/cytostatic activities (LD_%_) of the total supernatants derived from the toxicity assay displayed in Figure 1, instead of the concentrations of individual (partly unknown) toxins. Ch:Sm-liposomes provided full protection for all types of immune cell lines against the high, saturating the cytotoxic activities (LD_>90_) of both GAS supernatants (Figure 2a,b). This finding suggests that SLO is the major cytotoxin produced by GAS.

Only THP-1 cells were fully protected by Ch:Sm-liposomes from the cytotoxic effects of the GGS ATCC 12394 supernatant (Figure 2c). In addition to SLO, the GGS ATCC 12394 supernatant possessed a significant SLS hemolytic activity [21]. As THP-1 cells were fully protected against this supernatant by Ch:Sm-liposomes that are not active against SLS, this finding highlights their high resistance towards SLS. However, the SLS activity of this strain was readily detectable in experiments with Jurkat and Raji cells, and manifested itself in only partial protection, which was achieved by SLO-neutralizing Ch:Sm-liposomes (Figure 2c).

As expected, irrespective of the type of immune cell line, Ch:Sm-liposomes showed no or very little protection against GGS 5804 that secretes predominantly SLS, even when the supernatant was used at the non-saturating dose (LD_70–90_) to detect the minor SLO contribution (Figure 2d,e) [21].

Taken together, these results demonstrate that the protection efficiency of Ch:Sm-liposomes depends on the ratio between SLO and SLS secreted by individual bacterial strains, as well as on the individual sensitivity of different immune cells towards SLO and SLS. In addition, our data corroborate our previous findings that GGS relies more heavily on SLS for its cell membrane-damaging activity compared to mostly SLO-dependent GAS [21].

### 3.3. 18:1/18:1 PC-Liposomes Protect Raji and THP-1 but Not Jurkat Cells against SLS Secreted by GGS

We have shown earlier that liposomes composed exclusively of unsaturated lipids, such as 18:1/18:1 PC, neutralized SLS, but possessed no neutralizing activity against SLO [21].

Therefore, next, we tested the ability of 18:1/18:1 PC-liposomes to protect immune cell lines against GAS and GGS supernatants. To unmask minor SLS activities, the supernatants were used in their non-saturating amounts (LD_60–90_), otherwise at the saturating dose, the SLO activity could be sufficient to kill all cells [21].

As SLO-neutralizing Ch:Sm-liposomes fully protected all types of immune cell lines against GAS supernatants, it is only to be expected that SLS-neutralizing 18:1/18:1 PC-liposomes showed no or only minimal protection against these supernatants (Figure 3a,b).

Likewise, as the Ch:Sm-liposomes fully protected THP-1 cells against GGS ATCC supernatant (Figure 2c), 18:1/18:1 PC-liposomes showed no protection for THP-1 cells against this supernatant (Figure 3c). This finding also corroborates our initial findings that the THP-1 cell line is most resistant to the cytotoxic action of SLS. SLS-neutralizing 18:1/18:1 PC-liposomes partially neutralized the cytotoxic activity of the GGS ATCC supernatant against Raji cells (Figure 3c), which is in line with the partial protection observed under similar experimental conditions by SLO-neutralizing Ch:Sm-liposomes (Figure 2c).

18:1/18:1 PC-liposomes partially neutralized the cytotoxic activity of mostly the SLS-producing GGS 5804 supernatant against both THP-1 and Raji cells (Figure 3d), which is in line with the partial (albeit relatively small) protection observed under similar experimental conditions by SLO-neutralizing Ch:Sm-liposomes (Figure 2e).

Surprisingly, 18:1/18:1 PC-liposomes provided no protection to Jurkat cells against any of the GGS supernatants (Figure 3c,d), despite the only partial protection that was observed under similar experimental conditions by Ch:Sm-liposomes (Figure 2c,e).

### 3.4. A Combination of Ch:Sm and 18:1/18:1 PC-Liposomes Provides Superior Protection against GGS Strains

Ch:Sm-liposomes alone were capable of protecting all types of immune cells used in this study against the cytotoxic activities of GAS supernatants. For GGS strains, partial protection was observed either by Ch:Sm or by 18:1/18:1 PC-liposomes, with a notable exception of Jurkat cells, for which no protection by 18:1/18:1 PC-liposomes against both GGS strains was observed.

Therefore, next, a mixture of Ch:Sm and 18:1/18:1 PC-liposomes was applied to immune cells that were not fully protected against specific supernatants by either of the liposomes. In this experiment, a constant amount of Ch:Sm-liposomes sufficient to completely inhibit the SLO activity of the supernatants was applied, together with incrementally increasing amounts of 18:1/18:1 PC-liposomes. The combination of Ch:Sm and 18:1/18:1 PC-liposomes fully protected not only Raji, but surprisingly also Jurkat cells, against the GGS ATCC 12394 supernatant (Figure 4a).

Likewise, full protection was observed for THP-1 cells and Jurkat cells against the GGS 5804 supernatant (Figure 4b). For Raji cells, the protection against this supernatant by the liposomal cocktail did not reach 100% (Figure 4b), which might be indicative of an additional minor cytotoxic activity that is not neutralized by either liposome.

## 4. Discussion

Streptococcal pathogens represent a massive health burden, causing life-threatening conditions including sepsis and toxic shock syndrome [12,15]. Immune cells are key players to fight pathogens, but are also prime targets for bacterial virulence factors [24,25]. Their elimination allows the pathogens to successfully infect and harm its host [24,25].

Immune cells are particularly at risk against secreted pore-forming toxins. Even after successful antibiotic treatment, lysed bacteria release large numbers of cytolysins [36]. Cytolysins exploit the differences in the cell membrane composition between the pathogen and the host. They target mostly sphingomyelin (Sm), phosphatidylcholine (PC), and cholesterol (Ch), lipids that are rare or absent from bacterial membranes [37].

In this study, we show that liposomal nanotraps composed of a combination of Ch, PC, and SM neutralize the major cytolysins secreted by GAS and GGS pathogens. This antivirulence approach protects a monocytic cell line (THP-1), as well as lymphoid T and B cell lines (Jurkat and Raji, respectively).

Bacteria-free supernatants of GAS and GGS strains contain potent cytolysins that kill immune cells [24,25]. All streptococcal strains tested produced SLO and SLS, but their relative contribution was species- and strain-dependent. GAS ATCC 19165 and GGS ATCC 12394 produced both toxins in significant quantities. GAS 50362 produced almost exclusively SLO, whereas GGS 5804 produced mostly SLS [21].

Taking advantage of high SLO’s affinity for cholesterol, we used Ch:Sm-liposomes to neutralize this toxin [21]. We confirmed direct binding of SLO to Ch:Sm-liposomes by mass spectrometry and tested the protection capacity of those liposomes to prevent cytolytic damage of the streptococcal supernatant against immune cells. As SLO activity is prevalent in GAS strains, low amounts of Ch:Sm were sufficient to protect all of the tested cell types against these pathogens.

The contribution of SLS was more prominent in GGS [21]. As Ch:Sm-liposomes neutralize SLO, but not SLS, at those concentrations, GGS supernatants were only partially neutralized [21]. Only THP-1 cells were fully protected from GGS ATCC 12394 supernatant, most likely due to their higher SLS resistance compared to Jurkat and Raji cells. This indicates that the protective capacity of the liposomes is influenced by the toxin sensitivity of the targeted cell.

We next tried to neutralize SLS with liposomal nanotraps. We used 18:1/18:1 PC-liposomes, which specifically inhibit SLS, but not SLO [21]. As GAS strains rely mostly on SLO for their cytotoxic activity, 18:1/18:1 PC-liposomes did not provide any protection against these pathogens.

Against GGS supernatants, the protective effect of 18:1/18:1 PC-liposomes was cell type dependent. Raji and THP-1 cells were partially protected. Surprisingly, 18:1/18:1 PC-liposomes did not protect Jurkat cells.

To maximize immune cell survival, we tried to neutralize all cytolysins simultaneously. We used a nanotrap cocktail composed of cholesterol-containing liposomes (low amounts) and unsaturated PC-liposomes (high amounts) to protect immune cells against GGS supernatants. Surprisingly, the protective effect of the liposomal cocktail was greater than the arithmetic sum of its individual parts.

Pore formation causes cell homeostasis dysregulation and can lead to cell death, which can affect surrounding cells. For example, TNFα is produced by immune cell in response to PFT challenges or is released after lysis in its transmembrane form, and can trigger extrinsic apoptosis [29,38,39,40,41]. It is possible that immune cell reaction when challenged with a toxin can cause death of the surrounding cells and may be prevented only by neutralizing both toxins simultaneously. Sensitivity differences may be due to differences in the expression or susceptibility to such factors [42,43].

Raji cells challenged with the GGS 5804 supernatant were significantly but not completely protected. This could be due to another secreted virulence factor or insufficient liposome amounts to inhibit all SLS.

Due to a low affinity for SLS, the amounts of liposomes required for the complete neutralization were too high to be suitable for further in vivo research. Streptolysin S (SLS) is a small, non-immunogenic, peptide. Its exact mode-of-action is still not yet fully clarified, and is beyond the scope of the current study. However, if subsequent study were to unveil SLS exact targets, this would allow for the fine-tuning of the liposome composition, increasing SLS-liposomes affinity, and significantly reducing the amounts of liposomes required for the neutralization.

We observed that the immune cells displayed different sensitivities towards SLO and SLS. The SLO sensitivity (Jurkat > THP-1 > Raji) differed from the SLS sensitivity (Raji > Jurkat > THP-1). Cytotoxicity differences of SLO and SLS were shown for other cell types [44]. Different factors can potentially justify these differences. SLO was recently shown to interact with a mannose receptor on specific immune cells [45]. It is possible that SLS also interacts with a receptor, potentially differentially expressed on immune cell membranes. Pore size and Ca^2+^ permeability are determinant factors for the repair ability of pores [46,47]. SLO and SLS rely on different modes of action and most likely cause different forms of membrane damage [48]. Cells can use different strategies to mitigate the damage from pore-forming toxins. They can, for example, shed the damaged membrane region on microvesicles [23]. The repair efficacy varies between immune cell types. It has been shown that THP-1 cells shed more than Jurkat cells after pneumolysin challenge [23].

The increasing burden of antibiotic-resistant bacteria means fewer effective medications to treat a population getting more susceptible due to aging and an increased number of immuno-compromised patients [1]. Antivirulence drugs are a promising alternative as well as a complementary strategy. Their aim is to target and neutralize virulence factors to undermine the pathogen ability to establish an infection [4]. Exotoxins are ideal targets for such drugs, as the majority of pathogens secrete and rely on them for their virulence [3].

PFTs bind to exposed molecular components of the cell membranes. Liposomal nanotraps mimic the outer layer of the targeted cell membranes, enriched in the toxin’s targets. The diverted toxins attack the liposomes rather than the host cells. After toxin neutralization, the immune system is capable of fighting back the pathogen more efficiently. Processes essential for bacterial growth or survival are not impacted by antivirulence treatment, thereby limiting the selection for resistance mechanisms and leaving the host microbiota unharmed [4]. Liposomal nanotraps are constituted of naturally occurring lipids; they are not immunogenic or toxic [30].

PFTs are widely spread in diverse genus, including *Clostridium*, *Listeria*, *Bacillus*, and *Escherichia* [39]. Animals such as sea anemones, spiders, and snakes also produce PFTs in their venom [49]. As PFTs share common plasmalemmal targets, a single liposome or a simple liposomal mixture would be capable of neutralizing multiple toxins secreted by a variety of pathogens, unlike neutralizing antibodies, which are highly specific [29].

## Figures and Tables

**Figure 1 cells-11-00166-f001:**
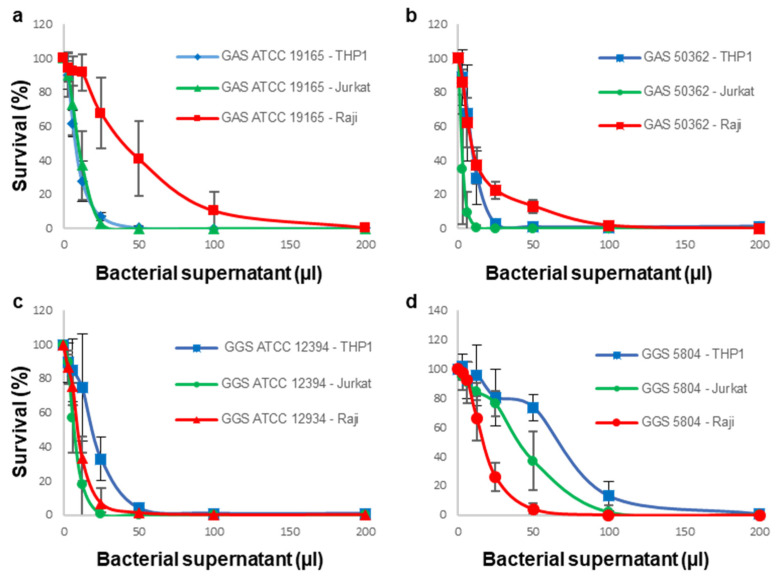
Different types of immune cells display different sensitivities to filtered bacterial supernatants. The GAS ATCC 19165 supernatant is more toxic to THP-1 and to Jurkat cells than to Raji cells (**a**). The GAS 50362 supernatant is more toxic to Jurkat cells compared to THP-1 cells; Raji cells are the most resistant (**b**). The GGS ATCC 12394 supernatant shows comparable toxicity towards Raji and Jurkat cells; THP-1 cells are the most resistant (**c**). The GGS 5804 supernatant is more toxic to Raji cells compared to Jurkat cells; THP-1 cells are the most resistant (**d**).

**Figure 2 cells-11-00166-f002:**
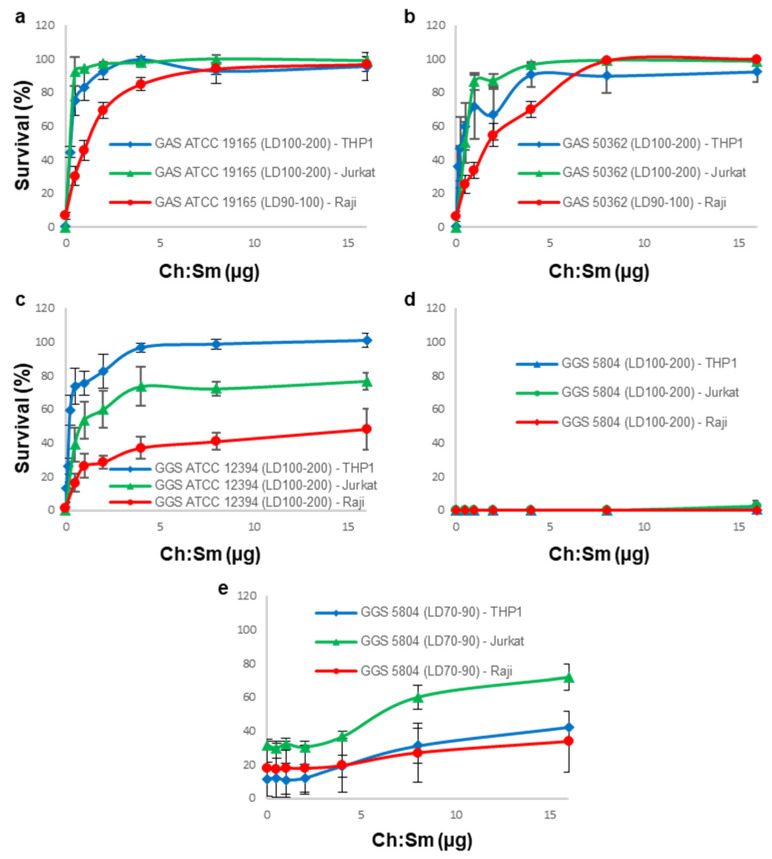
Neutralization of the cytotoxins secreted by GAS and GGS by cholesterol-containing liposomes. Ch:Sm-liposomes (Ch = 66 mol/%) completely neutralized cytolytic activities of GAS supernatants (LD_>90_) towards all immune cell lines used in the current study (**a**,**b**). Ch:Sm-liposomes provided full protection to THP-1 cells against the GGS ATCC 12394 supernatant (LD_>90_), but only partial protection for Jurkat and Raji cells. Protection levels differ significantly between all cell types, *p*-values < 0.0005 (**c**). Ch:Sm-liposomes provided no protection against the GGS 5804 supernatant (LD_>90_) for any cell type (**d**). Ch:Sm-liposomes provided partial protection against the GGS 5804 supernatant for THP-1 cells (~35% of ~LD_90_, *p*-value < 0.005), Jurkat cells (~40% of ~LD_70_, *p*-value < 0.008), and Raji cells (~15% of ~LD_85_, *p*-value < 0.005) (**e**). Error bars = mean ± SD; N ≥ 3.

**Figure 3 cells-11-00166-f003:**
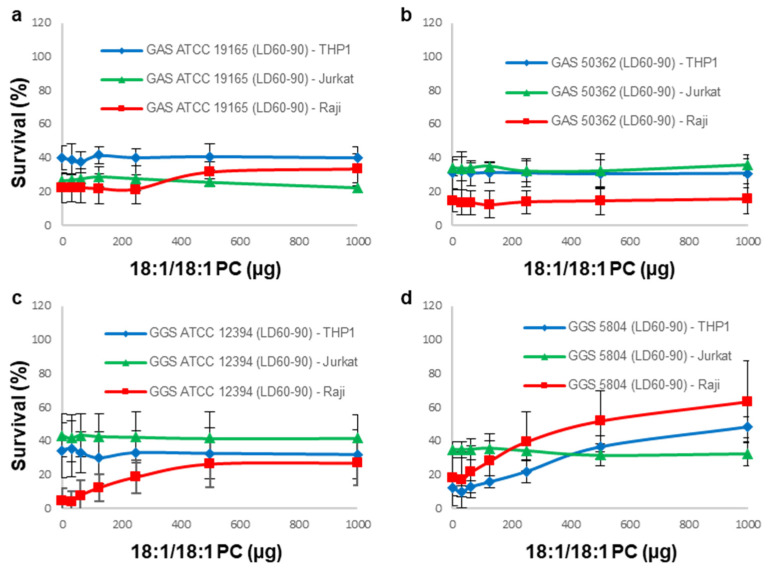
Inhibition of cytolysins secreted by GAS and GGS by liposomes composed of 18:1/18:1 PC. 18:1/18:1 PC-liposomes displayed no protection towards any immune cell line used in the current study treated with either of the GAS supernatants (**a**,**b**). No protection for THP-1 or Jurkat cell lines and partial protection for the Raji cell line were observed against the GGS ATCC 12394 supernatant (~20% of ~LD_95_, *p*-value < 0.03) (**c**). No protection for Jurkat cells and partial protection for THP-1 cells (~30% of ~LD_90_, *p*-value < 0.005) and Raji cells (~45% of ~LD_85_, *p*-value < 0.006) were observed against the GGS 5804 supernatant (**d**). Errors bars = mean ± SD; N ≥ 3.

**Figure 4 cells-11-00166-f004:**
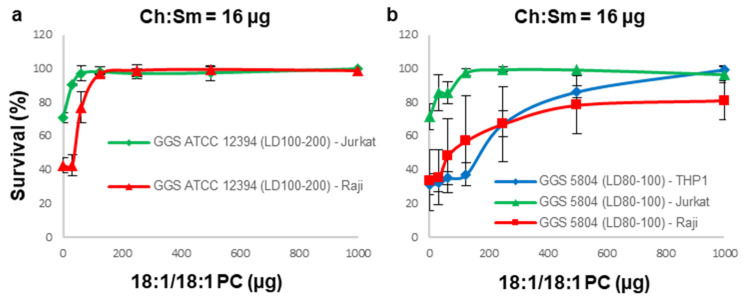
Inhibition of the cytolysins secreted by GGS by a liposomal mixture composed of Ch:Sm and 18:1/18:1 PC-liposomes. The combination of Ch:Sm-liposomes with 18:1/18:1 PC-liposomes completely protects immune cells against GGS ATCC 12394 cytolysins (**a**). Against GGS 5804 cytolysins, full protection is reached by THP-1 and Jurkat cells, but it is only partial for Raji cells (**b**).

## Data Availability

All data generated or analyzed during this study are included in this published article or are available upon request.

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
