# Peer review of "Engineered Liposomes Protect Immortalized Immune Cells from Cytolysins Secreted by Group A and Group G Streptococci†"

_cells, 2022, doi:10.3390/cells11010166_

Round 1

Reviewer 1 Report

In this manuscript, the Authors examined the efficacy of the liposome preparations to neutralise the exotoxins like SLO and SLS present in the bacterial culture medium. I have the following comments.

  1. The authors used the term liposome nanotraps. The authors have used conventional liposome preparations. It is not clear why they are calling the preparations as the nanotraps.
  2. The authors have used bacterial culture supernatants to examine the cytotoxic effect on the THP-1, Jurkat, and Raji cell lines. Quantitation fo the the culture medium is required (by measuring the total protein content, or by providing some quantitative indicator). Volume of the culture supernatant is mentioned in the results. It is not quantitative, and will vary from batch to bath of the bacterial culture.
  3. LDH-release assay of cytotoxicity or MTT assay of cell viability should have been used for measuring the cell death.
  4. It is well-known that SLO binds to the cholesterol-containing lipid bilayer. Therefore, it is obvious that the cholesterol-containing liposomes would neutralise SLO present in the bacterial culture. Therefore, no new information is provided with these experiments.
  5. It appears that the cholesterol-containing liposomes provide partial protection against the GGS supernatant. This is an interesting observation. However, authors did not make any attempt to explore this aspect in mechanistic detail.
  6. In my opinion, differential inhibition of the SLS activity in GGS supernatant by the liposome preparations is interesting, and may provide new insights, if explored in more detail.

Reviewer 2 Report

As clearly described by the authors in the introduction, liposomal nanotraps can neutralize exotoxins produced by various pathogens by becoming targets of these exotoxins initially directed at host immune cells.

The paper by Besançon et al. analyzes whether a mixture of cholesterol-containing liposomes (Ch:Sm-liposomes) with phosphatidylcholine liposomes (PC-liposomes) is protective against the combined action of SLO and SLS streptococcal exotoxins. To this end, they design a series of experiments in which they confront different monocytc and lymphoid cell lines to cytolysins in the absence and presence of the decoy liposomes.

The title and abstract are appropriate for the content of the text. The article is well constructed and the experiments were well conducted although cell survival experiments should be more detailed especially in terms of the results shown. I miss a concluding paragraph to replace some of the last paragraphs of the discussion, most of which are redundant with the information provided in the introduction.

Specific comments/questions

  • Although the sensitivity of immune cells to GAS or GGS supernatants is shown in the first section of the results, were any controls used in the neutralization assays with liposomes? 
  • In the material and methods section it is stated that the cells survival assays start with the addition of a fixed volume of supernatant but it is not clear to me the criteria for reflecting different amounts of supernatant in Figures 2 and 3. It should be explained more clearly. What is the reason for using different amounts of supernatant with cytolysins depending on its source or the type of cell line it is tested against in figures 2 and 3? This should be explained. Furthermore, it should be justified why this difference does not influence the comparison of the results with the different cell lines (THP1, Jurkat, Raji). 

Round 2

Reviewer 1 Report

I have no more comments.